# Effective Detection of Nafion^®^-Based Theranostic Nanocapsules Through ^19^F Ultra-Short Echo Time MRI

**DOI:** 10.3390/nano10112127

**Published:** 2020-10-26

**Authors:** Natalia Łopuszyńska, Krzysztof Szczepanowicz, Krzysztof Jasiński, Piotr Warszyński, Władysław P. Węglarz

**Affiliations:** 1Institute of Nuclear Physics, Polish Academy of Sciences, 31–342 Krakow, Poland; natalia.lopuszynska@ifj.edu.pl (N.L.); krzysztof.jasinski@ifj.edu.pl (K.J.); 2Institute of Catalysis and Surface Chemistry, Polish Academy of Sciences, 30–239 Krakow, Poland; ncszczep@cyf-kr.edu.pl (K.S.); ncwarszy@cyf-kr.edu.pl (P.W.)

**Keywords:** theranostic nanocarriers, polyelectrolites nanocapsules, imaging, ^19^F MRI, ultra short echo time, UTE

## Abstract

The application of the Three-Dimensional Ultra-Short Echo Time (3D UTE)pulse sequence at a high magnetic field for visualization of the distribution of ^19^F loaded theranostic core-shell nanocapsules with Nafion^®^ (1,1,2,2-tetrafluoroethene; 1,1,2,2-tetrafluoro-2- [1,1,1,2,3,3-hexafluoro-3-(1,2,2-trifluoroethenoxy)propan-2-yl] oxyethanesulfonic acid) incorporated into the shell is presented. The nanocarriers were formed via the layer-by-layer technique with biodegradable polyelectrolytes: PLL (Poly-L-lysine), and with Nafion^®^: polymer with high ^19^F content. Before imaging, an MR (magnetic resonance) spectroscopy and T_1_ and T_2_ measurements were performed, resulting in values of T_2_ between 1.3 ms and 3.0 ms, depending on the spectral line. To overcome limitations due to such short T_2_, the 3D UTE pulse sequence was applied for ^19^F MR imaging. First Nafion^®^ solutions of various concentrations were measured to check the detection limit of our system for the investigated molecule. Next, the imaging of a phantom containing core-shell nanocapsules was performed to assess the possibility of visualizing their distribution in the samples. Images of Nafion^®^ containing samples with SNR ≥ 5 with acquisition time below 30 min for ^19^F concentration as low as 1.53 × 10^−2^ mmol ^19^F/g of sample, were obtained. This is comparable with the results obtained for molecules, which exhibit more preferable MR characteristics.

## 1. Introduction

Since the results of the first study demonstrating the feasibility of fluorine magnetic resonance imaging (^19^F MRI) were published in 1977 [1], rapid advances in both MRI hardware and software allowed for the development and investigation of various applications of ^19^F MRI. The main research directions included the imaging of fluorinated drugs, cell tracking, inflammation imaging and cancer diagnostics, the developing of oxygen-sensing and -responsive probes, and designing targeted fluorinated imaging agents. Those applications were extensively described and their results were summarized in many reviews [2,3,4,5,6].

There are several preferable physical, chemical and biological properties making ^19^F MRI a potentially promising method for tracing theranostic agents—i.e., with combined therapeutic and diagnostic functionality. Gyromagnetic ratio of ^19^F nucleus is very close to the ^1^H value (40.08 vs. 42.58 MHz/T), which means that those spins process almost at the same resonance frequency (376.75 vs. 400.25 MHz at 9.4 T), allowing for the use of ^1^H MR hardware and software instrumentation with only a minimal adjustment for the ^19^F imaging [5]. Moreover, ^19^F has a natural abundance of 100%, resulting in nuclear magnetic resonance (NMR) receptivity (an overall NMR sensitivity of a nucleus at natural abundance) of 0.83 relative to ^1^H. The close to zero natural concentration of ^19^F nuclei in the human body makes fluorine atoms a perfect MRI marker without any natural background signal. This creates the opportunity of localizing and identifying only exogenous fluorinated compounds with 100% specificity of the signal, assuring excellent contrast-to-noise ratio. To track the distribution of ^19^F labeled therapeutics in an anatomical context, the ^19^F MRI results are overlaid over the conventional ^1^H image acquired in the same imaging session 

Nafion^®^ is a fluorinated polymer—high molecular weight, negatively charged molecule—which is relatively easy to incorporate it into nanocapsule shell via the layer by layer method [7,8]. From a synthesis point of view, this creates an excellent opportunity for designing a highly repetitive and efficient process for the production of theranostic nanocarriers for drug delivery. However, taking into consideration MR imaging, there are several issues making the visualization of the spatial distribution of Nafion^®^-containing nanocarriers in the sample challenging. Those challenges are complex, multipeak ^19^F NMR spectrum composed of relatively broad lines in order of 1–2 ppm which correspond to short (in range of single milliseconds) transverse magnetization decay.

The occurrence of multiple resonances in the ^19^F MR spectrum leads to chemical shift artifacts in ^19^F images, i.e., multiple, spatially shifted images in frequency encoding direction. While the simplest approach to avoid it is choosing molecules with multiple identical fluorine atoms, such as perfluoro-15-crown-5-ether (PFCE), using them it is not always straightforward due to their chemical properties. Alternatively, methods for compensation of unwanted resonance signals can be applied and, in some cases, if one of the resonances is separated enough from others, the compensation may be entirely omitted [9,10,11].

As many of the available ^19^F compounds exhibit short (in range of single milliseconds) transverse relaxation time, the choice of an adequate imaging sequence is of the utmost importance. There are only a few commercially available sequences allowing for the effective imaging of ^19^F fast-relaxing components. Those include the ultrashort echo time (UTE) or its 3D implementation (3D UTE) and the zero-echo time (ZTE) imaging sequences. Both sequences allow for the preservation of the signal from rapid-relaxing components, as they require very little time for preparation, maximizing the time for sampling [12,13]. Therefore, the optimization of imaging parameters of such sequences allows for the visualization of ^19^F compounds with less preferable characteristics.

The present work has aimed to assess the possibility to use 3D UTE pulse sequences (Figure 1) at a high magnetic field for visualization of the distribution of ^19^F loaded theranostic nanocapsules with Nafion^®^ molecules incorporated into the shell. The presented work and [8] present complementary parts of the research. While [8] is focused on nanocapsule synthesis and characterization, this paper provides the extensive description of MR imaging and spectroscopy experiments for both Nafion^®^ solution and Nafion^®^-loaded nanocapsules, which were accomplished in order to optimize the ^19^F MRI-based visualization of the nanocapsules distribution.

## 2. Materials and Methods

Nafion^®^ polymer (1,1,2,2-tetrafluoroethene;1,1,2,2-tetrafluoro-2-[1,1,1,2,3,3-hexafluoro-3- (1,2,2-trifluoroethenoxy)propan-2-yl] oxyethanesulfonic acid) is anionic polymer that is synthesized by the copolymerization of a perfluorinated vinyl ether comonomer with tetrafluoroethylene (TFE), resulting in the chemical structure given in Figure 2a [14,15].

The Nafion^®^ molecule has five fluorine nuclei groups that give rise to multiple signals in the MR spectrum. The structure of a Nafion^®^ unit illustrates the variability of the material—i.e., unreported by a vendor co-monomer distribution of molecular groups x and y that results in differences in ^19^F signal strength arising from a specified fluorine group from one synthesis to another.

The material used in experiments was Nafion^®^ 20 wt. % solution in lower aliphatic alcohols and water (663492 Sigma-Aldrich; Poznań, Poland)). The stock solution was first used to acquire a full ^19^F MR spectrum and to measure T_1_ and T_2_ relaxation times. Next, a series of Nafion^®^ dilutions in water with varied concentration were prepared to estimate the minimal number of ^19^F nuclei in a sample that can be visualized in reasonable scan time. Hexafluorobenzene purchased from Sigma Aldrich was used as a reference substance.

For theranostic nanocapsule synthesis, the polycation poly-L-lysine hydrobromide, PLL (MW 15 to 30 kDa) and polyanion: NAFION® (663492), as well as chloroform, sodium chloride and docusate sodium salt (AOT), were purchased from Sigma-Aldrich. All materials were used as received without further purification. The ultrapure water was produced using the Millipore Direct-Q5 UV Merck purification system.

All samples for MRI measurements were prepared in vials with micro-inserts containing a reference substance.

### 2.1. Polyelectrolyte Shell Liquid Core Nanocapsules Preparation and Characterization

The polyelectrolyte shell liquid core nanocapsules were prepared adopting the procedure proposed by us before [7]—i.e., by the encapsulation of nanoemulsion droplets in the polyelectrolyte multilayer shell. The details of the preparation procedure are given in [8]. Briefly, the oil phase for the nano emulsification was prepared by dissolving anionic surfactant AOT in chloroform at the concentration 340 g/L, while the water phase was prepared by dissolving poly-L-Lysine in 0.015 M NaCl solution (concentration varied from 10 to 300 ppm). The liquid core of capsules (nanoemulsion droplets) was formed by dispersing oil phase (0.1 mL) into the water phase (200 mL of polycation solution) during mixing with a magnetic stirrer at the rate 300 rpm. On such prepared liquid cores, the multilayer shell was formed by the subsequent adsorption of polyelectrolytes. After preparation, chloroform was evaporated with the final CHCl_3_ concentration not exceeding 0.04 mg/L [16].

### 2.2. MRI Equipment

Both ^19^F MR spectroscopy/relaxation measurements, as well as ^1^H and ^19^F 3DUTE imaging, were performed at the 9.4 T Bruker Biospec 94/20 research MRI scanner (Bruker Biospin, Ettlingen, Germany) with 210 mm bore diameter and high performance actively shielded BGA 12S HP gradient system (675 mT/m) with integrated shims. A small transmit-receive ribbon solenoid radiofrequency (RF) coil (ID of 14 mm), which can be tuned either to ^1^H or ^19^F resonant frequency (i.e., 400.130 vs. 376.498 MHz) was built and used for all experiments. Coil geometry was adjusted individually to analyzed sample size and shape to maximize the filling factor and thus Signal-To-Noise Ratio (SNR) values. Paravision 5.1 and Topspin 2.0 software was used to accomplish MR imaging and spectroscopy.

### 2.3. ^19^F Spectroscopy and Relaxometry

Prior to the imaging, an MR spectroscopy (with a 17 μs 90° pulse and SW: 59.52 kHz) and T_1_ and T_2_ measurements were performed in order to choose the peak with the largest area and consequently the highest NMR signal and to set imaging sequence parameters to values enabling an optimal visualization of Nafion^®^ molecule distribution.

The T_1_ measurement was performed in TopSpin 2.0 (Bruker Biospin, Ettlingen, Germany) using the pseudo-2D version of a standard Inversion Recovery experiment exploiting an array of 10 different inversion recovery (IR) experiments with different inversion time values (0.15, 0.30, 0.75, 1.5, 3.0, 6.0 s). T_2_ values for individual resonances were estimated by calculating the reciprocal of their full width at half maximum (FWHM) values.

### 2.4. 3D UTE Pulse Sequence

Conventional MRI pulse sequences generate relatively long echo times (>> 1ms). Therefore, the majority of the signal from short transverse relaxation time components decays to near zero before echo formation or even during RF pulse excitation. The 3D Ultrashort TE sequence allows for measuring very short T_2_ compounds, because of non-selective RF excitation pulse [17,18], and the minimum echo time (TE) is limited only by RF pulse duration and the time necessary to change between RF excitation and acquisition mode. A timing diagram for a 3D UTE sequence is presented in Figure 1. Sampling is performed from the start of the rising gradient ramp; thus, it begins always in the center of a k-space and continues to the surface of a sphere. The image reconstruction is performed by regridding from radial k-space on a Cartesian grid followed by a conventional 3D fast Fourier transformation.

### 2.5. 3D UTE Imaging

The application of 3D UTE to actual experiments requires performing a preparation specific for this method, including gradient delay optimization and k-space trajectory measurement. The optimal gradient delay was determined using a glass sphere phantom filled with an aqueous solution of CuSO_4_ positioned in the isocenter of the scanner. Imaging with a gradient delay value of 1 µs resulted in the most homogenous image with the least apparent artifacts. The k-space trajectory was measured using the same phantom and stored to be used during actual imaging experiments. As the measured trajectory is valid only for specific scanning parameters (given matrix size, field of view, slice orientation, and acquisition bandwidth), thus this adjustment was repeated for each FOV and matrix size planned to be used in an imaging experiment.

The 3D UTE sequence was first applied to Nafion^®^ solutions of a different concentration. Imaging parameters were as follows: repetition time (TR): 8 ms, TE: 0.16 ms, RF pulse bandwidth (BW): 4.27 kHz, FOV: 4.0 × 4.0 × 4.0 cm^3^. For ^1^H images: matrix size ( MTX): 128 × 128 × 128 and number of averages (NA):: 1, resulting in total acquisition time ∼6m 51s, while for ^19^F images: MTX: 32 × 32 × 32, NA: 64 and acquisition time: ∼27 m6 s. The flip angle (FA) was set to 6.4° according to the modified Ernst angle formula to account for the effects of a transverse relaxation during the RF excitation [3].

Subsequently, the imaging of a phantom containing theranostic nano-carriers was performed with imaging parameters, as follows: TR: 8 ms, TE: 0.16 ms, FA: 6.4°, RF pulse BW: 4.27 kHz, FOV: 4.0 × 4.0 × 4.0 cm. For ^1^H images: MTX: 128 ×128 × 128 and NA: 1, acquisition time ∼ 6 m 51 s, while for ^19^F images: MTX: 32 × 32 × 32, NA: 256 or 500 and acquisition time: ∼ 1 h 48 min 24 s and ∼ 3 h 31 m 44 s, respectively.

## 3. Results

The theranostic nanocapsules with liquid core and polyelectrolyte shells composed with PLL and Nafion were prepared using the layer by layer method and a saturation technique. The details of the preparation and characterization procedure are given in [8]. For the present study, nanocapsules containing two Nafion^®^ layers were chosen. The sizes of those theranostic nanocapsules measured by the dynamic light scattering method were ~150 nm. The final particle concentration determined by nanoparticle tracking analysis was ∼ 1.5 × 10^12^ nanocapsules/mL [8]. 

### 3.1. ^19^F MR Spectroscopy

Figure 2 shows the example of obtained ^19^F spectrum of NAFION® with five visible resonances. These signals were assigned to chemical groups containing equivalent fluorine nuclei. T_1_ and T_2_ relaxation times of individual groups, as well as half-widths of spectral lines and chemical groups assignment, are presented in Table 1. To estimate the number of ^19^F nuclei contributing to each resonance, 10 µL C_6_F_6_ was used as a reference.

The structure of a Nafion^®^ unit is shown on the inset in Figure 2a. To estimate the minimum concentration of Nafion^®^, giving a sufficient amount of signal to perform imaging on a single peak, the MR spectroscopy of a series of solutions, with decreased Nafion^®^ concentrations, was performed. Basing on a difference in 42 ppm peak area for subsequent concentrations, the estimation of a number of ^19^F nuclei in 1 ml of the sample giving a signal to the image was made. Using equivalent weight formula EW = 100x + 446 [14], and EW value 1100 g/mole, we calculated the average x length in the Nafion^®^ molecule (Figure 2a). Assuming the average length of x chain was 6.6, the approximate molar concentration of Nafion^®^ was calculated (Table 2).

To check if the incorporation of Nafion into nanocapsules alters the ^19^F spectrum, the MR spectroscopy was also performed for nanocapsules. Results were compared with the spectrum obtained for the Nafion solution and no significant shifts of peaks positions were observed (Figure 3). However, the peak broadening (between 1.5 to 2 times for the main peaks at 42 ppm and 0 ppm, respectively) is observed for sample with nanocapsules which correspond to faster decaying free induction decay (FID) (T_2_ in order 1–1.5 ms) available for imaging.

### 3.2. MR Imaging

In order to obtain good quality images, with the least apparent artifacts, a peak at +42 ppm was chosen for imaging experiments due to the relatively high MR signal and being well separated from the other resonances. Results of MR imaging are shown in Figure 4. SNR was calculated as a quotient of the mean signal intensity in the area of interest and the mean signal intensity measured with an entirely attenuated (150.00 dB) RF pulse. For a sample with an initial 20% Nafion^®^ solution, SNR value for NA = 64 was higher than 600; therefore, imaging with only one average would still produce an image with SNR in order of 80. For the lowest analyzed concentration, imaging within a reasonable acquisition time of less than 30 min produced an image with SNR = 5, which is still sufficient to reliably determine the spatial distribution of ^19^F nuclei in a sample.

Next, the imaging of a phantom containing core-shell nanocapsules with Nafion^®^ incorporated in one of the shell layers was performed to check the possibility of visualizing the distribution of complete theranostic carriers in samples.

The 3D UTE imaging of theranostic nanocapsules with total acquisition time ∼ 1 h 48 min (256 averages) gave SNR = 12, which was high enough for visualization of the ^19^F distribution (see [8]). Increasing the number of averages to 500 (acq. time: ∼ 3 h 32 min) caused improvement in SNR to ∼ 20 (Figure 5), which also improves the visualization of finer details of the ^19^F distribution, as illustrated in Figure 5.

## 4. Discussion

The most frequently reported substances in ^19^F MRI studies are perfluorocarbons (especially nanoemulsions of PFOB and PFCE). Depending on the chosen PFC and an MRI system that is used, the estimated number of ^19^F nuclei that can be visualized in a scan time below 1 hour is in the range of ∼10^16^ of ^19^F nuclei/voxel [19,20]. Nonetheless, different chemical forms, such as liposomes, micelles or ^19^F modified implants, have also been investigated. The application of liposomal structures for drug delivery was shown by Bo et al. [21]. The authors synthesized Doxorubicin (DOX)-loaded liposomes with high ^19^F content for use as ^19^F detectable drug-delivery at the therapeutic dose. They obtained a structure that resulted in a single ^19^F resonance line and a detection limit as low as 5 mmol of ^19^F for a scan time = 6 m 24 s. Other studies using various ^19^F copolymers for application as ^19^FMRI detectable tracers (spin-echo imaging with TE = 4.8 ms, TR = 3000 ms, and total imaging time restricted to 80 min) allowed the authors to obtain images with SNR values in the range from 3.9 to 6.9 (for samples with 2.3–9.3 mmol ^19^F/g concentration) [22]. Yet, the total scan time of that range is one of the limiting factors to introduce those substances to clinical applications.

As the NMR signal is proportional to field strength, high field systems provide higher SNR than typical clinical scanners enabling the imaging of ^19^F nuclei at very low concentrations. It is desirable that tracers should be characterized by the high fluorine content, simple ^19^F NMR spectrum, with a single, sharp, and intense peak, short T_1_ and long T_2_ [23]_._ Nevertheless, due to advances in MRI instrumentation, and ultrashort echo pulse sequence development, the range of fluorine compounds available as a new MRI marker can be expanded, including one with less optimal characteristics, but easier to incorporate into theranostic nanocapsules. The use of high field equipment in preclinical imaging, as well as a tendency towards increasing the magnetic field of clinical MRI scanners, pave the way for use of a broader range of ^19^F compounds as a tracer.

As the signal from ^19^F compounds is usually only a little above the detection limits, the low SNR is the biggest challenge in ^19^F imaging. To achieve a sufficient SNR value, a compromise between spatial resolution and total scan time must be made. Collecting the signal with multiple averages causes an increase in signal intensity by the factor of $\sqrt{NA}$, but at the same time, total scan time scales directly with NA. Another option to increase SNR is sampling a larger volume, which results in loss of spatial resolution. However, the use of a high-resolution ^1^H MR image as an anatomical framework for the presentation of the ^19^F-compound distribution alleviates this inconvenience.

In the presented work we managed to obtain ^19^F MR images of polyelectrolyte theranostic nanocapsules containing Nafion^®^ polymer as a tracer, with SNR ≥ 5 and an acquisition time below 30 min, for ^19^F concentration as low as 1.53 × 10^−2^ mmol ^19^F/g of sample. This is comparable with the results obtained for molecules that are theoretically more suitable for application as ^19^F MRI tracers.

In our previous work, the effective contrasting properties of iron oxides and gadolinium-based compounds were effectively tested for the visualization of the distribution of the polyelectrolyte theranostic nanocapsules, using ^1^H MR imaging [24,25]. The possibility to use the ^19^F-containing Nafion^®^ compound, which is relatively easy to incorporate into a nanocapsule’s layers, broadens the potential applicability of these theranostic nanocarriers. The use of a 3D UTE pulse sequence at a high magnetic field of 9.4T allows for effective imaging of the presence of nanocapsules with Nafion^®^ incorporated into their shell. This indicates the possibility to use this approach in preclinical research, for the potential imaging of the in vivo distribution of nanocarriers, without a background signal, as is in the case with ^1^H MRI.

## 5. Conclusions

We showed that the application of the 3DUTE sequence at high field makes heteronuclear UHF MRI a promising tool for the assessment of theranostic nanocarriers distribution, even in the case of molecules without a single, narrow ^19^F spectrum line. We obtained ^19^F MR images of the Nafion^®^ polymer containing samples with SNR ≥ 5 and with an acquisition time below 30 min for ^19^F concentrations as low as 1.53 × 10^−2^ mmol ^19^F/g of a sample. This is comparable with the results obtained for molecules that are theoretically more suitable for applications as ^19^F MRI tracers. The obtained results show the potential possibility to use this approach for the assessment of nanocarrier distribution in preclinical research or clinical applications performed at a high magnetic field MRI.

## Figures and Tables

**Figure 1 nanomaterials-10-02127-f001:**
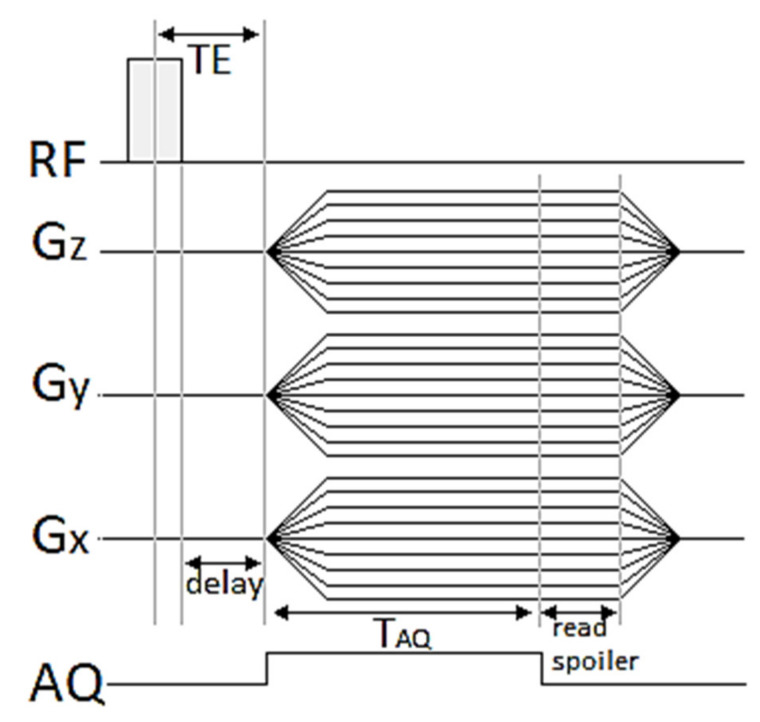
Three-Dimensional UTE Imaging pulse sequence diagram.

**Figure 2 nanomaterials-10-02127-f002:**
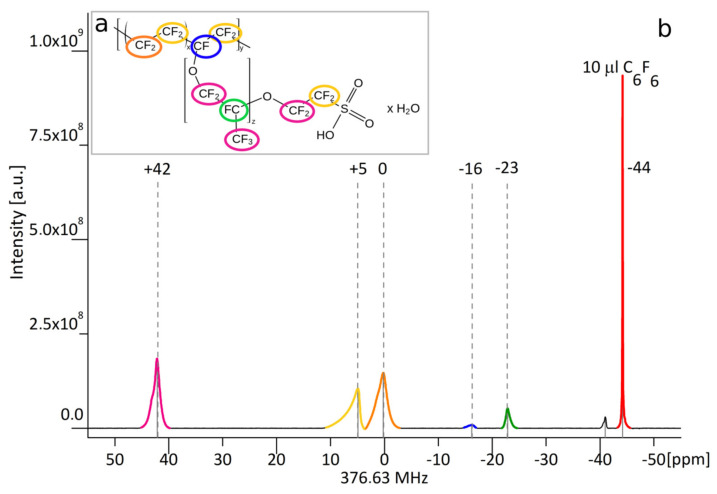
(**b**) ^19^F MR spectrum of original Nafion^®^ used as a source for further sample preparation with C_6_F_6_ used as a reference. Different colors correspond to individual chemical groups, as marked on chemical structure (**a**).

**Figure 3 nanomaterials-10-02127-f003:**
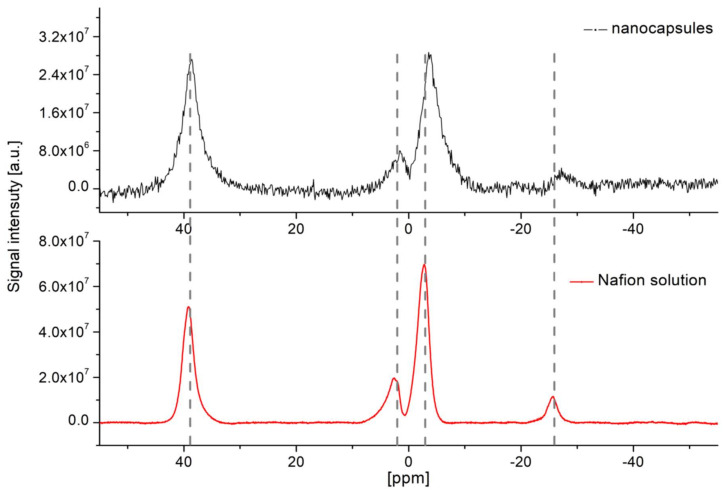
Comparison of acquired ^19^F MR spectra of Nafion^®^ loaded nanocapsules (NA = 5800) and solution (sample N1, NA = 256). The corresponding fluorine concentrations (in terms of the number of ^19^F molecules per 1 mL of the sample) were 2.8 × 10^18^ and 1.5 × 10^20^, respectively.

**Figure 4 nanomaterials-10-02127-f004:**
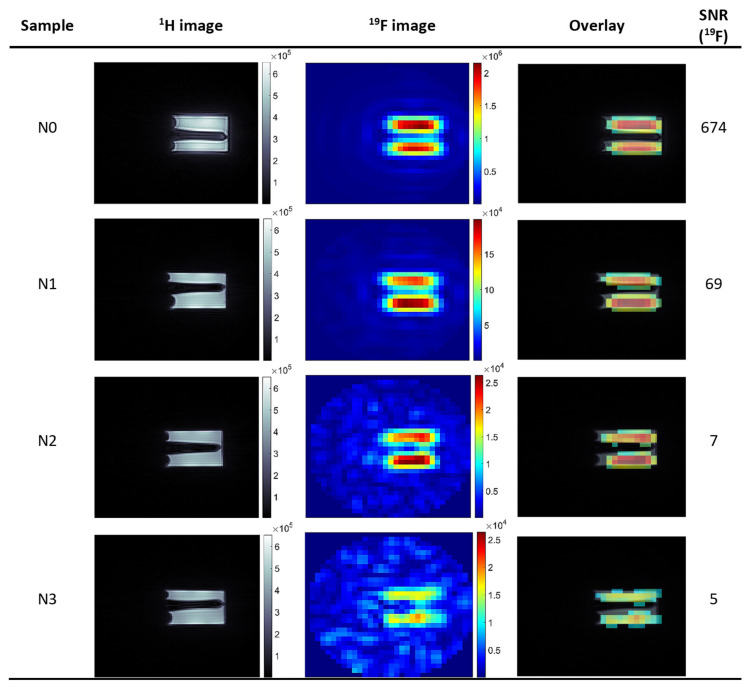
MR Imaging results. Series of ^1^H and ^19^F axial images with absolute intensities as displayed on bars. First column ^1^H MR Images of Nafion^®^ solutions of a different concentration; second column: corresponding ^19^F MR images and (third column) an overlay of ^1^H and ^19^F images. Parameters of UTE3D sequence TR: 8 ms, TE: 0.16 ms, FA: 6.4°, RF pulse length: 0.3 ms, FOV: 4.0 cm, NA:1 or 64 (for ^1^H or ^19^F) MTX: 128 × 128 × 128 or 32 × 32 × 32 (^1^H/^19^F), total acquisition time: ∼ 6 m 51 s/27 m 6 s(^1^H/^19^F).

**Figure 5 nanomaterials-10-02127-f005:**
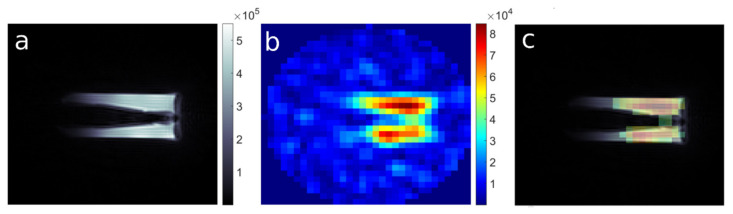
(**a**) ^1^H MR Image of a phantom containing nanocapsules (UTE3D, FOV: 4.0 cm, MTX 128, NA:1), (**b**) corresponding ^19^F MR image (UTE3D, FOV: 4.0 cm, MTX: 32, NA: 500) and (**c**) an overlay of ^1^H and ^19^F images.

**Table 1 nanomaterials-10-02127-t001:** Observed resonances with chemical groups assignment and calculated T_1_ and T_2_ relaxation times of individual peaks.

Chemical Shift (ppm)	Corresponding Chemical Groups	FWHM (Hz)	T_1_ (ms)	T_2_ (ms)
+42	OCF_2_, OCF_2_, CF_3_	483	909	2.1
+5	CF_2_, CF_2_, SCF_2_	739	691	1.4
0	(CF_2_)_a_	789	733	1.3
−16	CF (II)	640	775	1.6
−23	CF (I)	329	864	3.0

**Table 2 nanomaterials-10-02127-t002:** Estimated number of ^19^F nuclei in 1 ml of the sample based on peak area changes.

Sample	Relative Concentration of 20% Nafion^®^ Solution in H_2_O (mL)	Number of ^19^F Nuclei in 1 mL of the Sample Giving a Signal in + 42 ppm Peak	Total Number of ^19^F Nuclei in 1 mL of the Sample	Estimated Concentration (mM)
N0	1.000	6.4 × 10^20^	2.1 × 10^21^	88.50
N1	0.100	4.5 × 10^19^	1.5 × 10^20^	6.32
N2	0.010	6.0 × 10^18^	2.0 × 10^19^	0.84
N3	0.005	2.8 × 10^18^	9.2 × 10^18^	0.39

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
