# Peer review of "Effective Detection of Nafion®-Based Theranostic Nanocapsules Through 19F Ultra-Short Echo Time MRI"

_nanomaterials, 2020, doi:10.3390/nano10112127_

Round 1
Reviewer 1 Report
The manuscript by Weglarz et al described the application of UHF MRI for theranostic diagnosis with 3DUTE sequence at high field. Overall the manuscript is well structured and written. Authors addressed adequate explanation for the study resluts and provided insightful understanding of UHF MRI application for biomedical filed. According to its current shape, I recommend it published in nanomaterials.
Author Response
We thank the reviewer for appreciating our work. We did some improvement of the paper as an answer to the direct comments of the reviewer nr 3.
Reviewer 2 Report
see attached file

Author Response
We thank the reviewer for appreciating our effort.
Reviewer 3 Report
The manuscript by Lopuszynsk et al. shows the logical steps taken to assess the possibility to detect Nafion-nanocapsules using 19F 3DUTE MRSI within sensible timeframes. However, the aims, conclusions, and figures of 19F 3DUTE using exactly the same parameters are already covered within their previous publication earlier this year (Ref [8]: Szczech et al. 2020, Langmuir). This needs to be addressed and altered, if this is possible, in order to make the manuscript novel.
It is good to see Figure 1, which I didn't notice in the previous paper. If the limiting factor is enough SNR, then would it make sense to find the T1s when loaded within nanocapulses, and possibly also within in ex vivo tissue? I know you have aimed for the Ernst angle, but is there an optimal TR/Ernst angle combination that optimises SNR?
Minor points:
- Figure ordering/numbering appears awry.
- Fig 2 comparing Nafion solution with nanocapsules. You mention the NA, but what were the concentrations? Ie. It would be nice to compare SNR as well as lineshape/frequency differences.
Round 2
Reviewer 3 Report
.